

# Prediction of endoplasmic reticulum resident proteins using fragmented amino acid composition and support vector machine

Ravindra Kumar[1,2], Bandana Kumari[1] and Manish Kumar[1]

[1] Department of Biophysics, University of Delhi South Campus, New Delhi, India
[2] Current affiliation: Newe-Ya'ar Research Center, Agricultural Research Organization, Ramat Yishay, Israel

## ABSTRACT

**Background**. The endoplasmic reticulum plays an important role in many cellular processes, which includes protein synthesis, folding and post-translational processing of newly synthesized proteins. It is also the site for quality control of misfolded proteins and entry point of extracellular proteins to the secretory pathway. Hence at any given point of time, endoplasmic reticulum contains two different cohorts of proteins, (i) proteins involved in endoplasmic reticulum-specific function, which reside in the lumen of the endoplasmic reticulum, called as endoplasmic reticulum resident proteins and (ii) proteins which are in process of moving to the extracellular space. Thus, endoplasmic reticulum resident proteins must somehow be distinguished from newly synthesized secretory proteins, which pass through the endoplasmic reticulum on their way out of the cell. Approximately only 50% of the proteins used in this study as training data had endoplasmic reticulum retention signal, which shows that these signals are not essentially present in all endoplasmic reticulum resident proteins. This also strongly indicates the role of additional factors in retention of endoplasmic reticulum-specific proteins inside the endoplasmic reticulum.

**Methods**. This is a support vector machine based method, where we had used different forms of protein features as inputs for support vector machine to develop the prediction models. During training *leave-one-out* approach of cross-validation was used. Maximum performance was obtained with a combination of amino acid compositions of different part of proteins.

**Results**. In this study, we have reported a novel support vector machine based method for predicting endoplasmic reticulum resident proteins, named as ERPred. During training we achieved a maximum accuracy of 81.42% with *leave-one-out* approach of cross-validation. When evaluated on independent dataset, ERPred did prediction with sensitivity of 72.31% and specificity of 83.69%. We have also annotated six different proteomes to predict the candidate endoplasmic reticulum resident proteins in them. A webserver, ERPred, was developed to make the method available to the scientific community, which can be accessed at http://proteininformatics.org/mkumar/erpred/index.html.

**Discussion**. We found that out of 124 proteins of the training dataset, only 66 proteins had endoplasmic reticulum retention signals, which shows that these signals are not an absolute necessity for endoplasmic reticulum resident proteins to remain inside the endoplasmic reticulum. This observation also strongly indicates the role of additional

Corresponding author
Manish Kumar,
manish@south.du.ac.in

factors in retention of proteins inside the endoplasmic reticulum. Our proposed predictor, ERPred, is a signal independent tool. It is tuned for the prediction of endoplasmic reticulum resident proteins, even if the query protein does not contain specific ER-retention signal.

## INTRODUCTION

The endoplasmic reticulum (ER) is an important organelle of eukaryotic cells. It participates in several essential cellular activities, such as quality control of protein production, recognition of mis-folded proteins, lipid biosynthesis, detoxification, steroids and xenobiotic compound metabolism, protection against cellular stress, intracellular calcium homeostasis, and intracellular signalling (*Lavoie & Paiement, 2008*; *Verkhratsky, 2002*). ER malfunction might lead to many diseases like cystic fibrosis, cancer and juvenile pulmonary emphysema (*Robinson-Rechavi et al., 2001*) and neurological disorders like ischemia and epileptic seizures (*Paschen & Frandsen, 2001*). ER is involved in several co-translational and post-translational modifications like signal-peptide cleavage, disulphide bond formation, N-linked glycosylation, and glycosylphosphatidylinositol (GPI)-anchor formation (*Barz & Walter, 1999*) and is also essential for post-translational processing of secretory proteins that makes it one of the important components of eukaryotic secretory protein system (*Barlowe & Miller, 2013*). Secretory proteins are synthesized on rough ER and routed across the ER membrane into lumen through a co-translational process. Inside the ER, they undergo glycosylation and attain their specific 3D conformation before being further transported downstream in the secretory system. The whole process is achieved with the help of highly coordinated action of several proteins which includes molecular chaperones and different enzymes (*Nakatsukasa & Brodsky, 2008*). Proteins, which help ER in carrying out above-mentioned functions, do not get secreted, and they are called as ER resident proteins (ERRPs). In addition to ERRPs, mis-folded or unfolded proteins are also retained back in ER and forwarded for proteasome-mediated degradation if the error could not be rectified (*Sitia & Braakman, 2003*; *Van Anken & Braakman, 2005*; *Wang & Hebert, 2003*). Therefore, in order to remain localized in ER, the ERRP must be recognized precisely and discriminated from the misfolded proteins or proteins of secretory pathway.

The function of ER is intricately bound with the Golgi apparatus. For example, similar to ER, from Golgi apparatus, also the proteins automatically proceed downstream to the plasma membrane or vacuoles in absence of an active retention process. Also a number of ERRPs are forced to localize in ER by continuous retrieval from the Golgi complex. Luminal ER proteins may be retrieved by using retrograde transport with help of C-terminal KDEL sequence while Type I transmembrane proteins—including ER residents and itinerant proteins of the ER/Golgi system—contain a C-terminal dilysine motif that signals their retrieval from post-ER membranes (*Gaynor et al., 1994*; *Jackson, Nilsson & Peterson, 1993*;

*Townsley & Pelham, 1994*). The dilysine motif known as KKxx and KxKxx motifs, consists of a pair of lysine residues at the C terminus of the cargo protein (*Jackson, Nilsson & Peterson, 1990*; *Nilsson, Jackson & Peterson, 1989*). Many variants of the dilysine motif have been reported such as KxHxx retrieval motif in the tail of the spike protein of group I coronaviruses (*Lontok, Corse & Machamer, 2004*), and the RKxx motif present in the Golgi-localized Scyl1 protein (*Burman et al., 2008*).

In past, a number of studies have focused on identification of proteins located in different subcellular organelles like nucleus (*Brameier, Krings & MacCallum, 2007*; *Huang et al., 2007*; *Kumar & Raghava, 2009*), mitochondrion (*Guda, Fahy & Subramaniam, 2004*; *Kumar, Verma & Raghava, 2006*), chloroplast (*Emanuelsson, Nielsen & Von Heijne, 1999*), Golgi body (*Chou, Yin & Xu, 2010*), peroxisome (*Emanuelsson et al., 2003*; *Neuberger et al., 2003*) but ERRPs remain unexplored. Despite the fact that the functionality of Golgi and ER are intricately intertwined with each other, several Golgi prediction methods have been developed (*Chou, Yin & Xu, 2010*; *Jiao & Du, 2016*). No attempt has been made to develop species neutral ERRPs predictor. Though in few subcellular localization prediction methods, ER has been included as a location, but at present, to the best of our knowledge, a predictor which can specifically predict ERRPs does not exist. Moreover, predictors like ProLoc-GO (*Huang et al., 2008*), KnowPred$_{site}$ (*Lin et al., 2009*), SLocX (*Ryngajllo et al., 2011*), iLoc-Animal (*Lin et al., 2013*), iLoc-Euk (*Chou, Wu & Xiao, 2011*), Cello v-2.5 (*Yu et al., 2006*), HybridGO-Loc (*Wan, Mak & Kung, 2014*), mGOASVM (*Wan, Mak & Kung, 2012*), Hum-Ploc (*Chou & Shen, 2006*), Euk-mPloc (*Chou & Shen, 2007*), PSLT (*Scott, Thomas & Hallett, 2004*), Euk-mPloc 2.0 (*Chou & Shen, 2010*), the method developed by *Cherian & Nair (2010)*, and *Guo et al. (2016)* considered ER as one among many subcellular locations. But these have some shortcomings like (i) among the above mentioned predictors, none were designed specifically to predict ERRPs; (ii) datasets used for training for prediction model were very old; (iii) subcellular locations were determined for a particular organism or groups (plant/animal/viral); (iv) many of them do not provide webserver/standalone software for scientific purpose and if some of them does so, they are not in working condition.

We believe that identification of all ERRPs is necessary to explore more about their functional associations and for better understanding of ER functional machinery. In the present study, we report a dedicated method, ERPred, to predict ERRPs with 81.42% accuracy. ERPred is based on the support vector machine (SVM) and the individual amino acid compositions of 25 N-terminal, 25 C-terminal and remaining amino acids as SVM input. We have also developed a freely accessible webserver and software that predicts ERRPs using their amino acid sequence. We hope that the present work will help in better understanding of cellular secretory pathways as well as other ER-mediated functions.

## MATERIALS AND METHODS

### Training datasets

Dataset compilation is one of the most important steps during development of a prediction method. To avoid any ambiguity, the dataset that has to be used for training the predictor

**Table 1  Distribution of ER resident and non-resident proteins in different datasets.**

| Proteins | Training dataset | Independent dataset |
|---|---|---|
| Endoplasmic Reticulum Resident Proteins | 124 | 65 |
| Non- Endoplasmic Reticulum Resident Proteins | 1,200 | 2,900 |

should contain experimentally annotated proteins of high quality. We found that there exists a database of human ER proteins (both known and potential) named as HERA (*Scott et al., 2004*). Since HERA is not updated and contains only human proteins hence we selected SwissProt, the manually curated section of UniProt protein database, to download the data. To retrieve the ERRPs from SwissProt, we used following criteria (i) proteins should be of eukaryotic origin and reviewed; (ii) should be present only in ER; (iii) existence proven by 'evidence at protein level'; (iv) protein should be full length and not fragmented (v) should have more than 50 amino acids because smaller proteins are mostly fragments; and (vi) protein location should be experimentally verified. For negative dataset, we tried to collect all types of non-ERRPs so that the predictor is trained on a comprehensive dataset and it can easily differentiate ERRPs from the non-ERRPs. We selected two different types of proteins as non-ERRPs (1) proteins which are N-glycosylated, since N-glycosylation occurs in ER (*Bieberich, 2014*; *Roth et al., 2010*); (2) non N-glycosylated proteins, since not all ER proteins are N-glycosylated. Both types of non-ERRPs were downloaded using the criteria earlier used to download ER proteins.

Proteins downloaded for making dataset may contain similar or homologous sequences hence if they are used without reducing the redundancy among proteins, the obtained performance might not be a genuine performance but an over-estimation. Hence most prediction methods have adopted the strategy of homology-reduced dataset to assess the performance of a predictor. In this work the redundancy was reduced to 40% using CD-HIT (*Li & Godzik, 2006*) and 124 ERRPs were obtained which were used as positive dataset. 1,200 eukaryotic non-ERRPs were used as the negative dataset (Table 1; Table S1). The 1,200 non-ER proteins were 10 times the positive dataset, which in our view should be sufficient for the predictor to model non-ERRPs (*Kumar et al., 2015*).

## Independent datasets

To assess the unbiased performance of a newly developed method, the evaluation must be carried out on a dataset which was not used during the training. So we created an independent dataset for this purpose. This dataset also contains both ERRPs and non-ERRPs. The source of ERRPs was also SwissProt (Release: 2015_06) and contained proteins, which had existence at 'evidence at transcript level', 'inferred from homology' and 'predicted'. Following criteria were used to download the ERRPs: (i) location should be endoplasmic reticulum; (ii) protein length should be greater than 50 amino acids; (iii) protein should be non-membranous; (iv) proteins should not be experimentally verified. After removing sequences, which had a redundancy of more than 40% among independent or with the training dataset, finally we got 65 proteins (Table 1; Table S2).

The non-ERRPs were also downloaded from SwissProt using following filters: (i) protein length should be greater than 50 amino acid; (ii) protein should be of eukaryotic origin and

non-membranous; (iii) protein's existence should be experimentally verified; (iv) removed the low quality annotation by excluding sequences annotated as 'by similarity', 'fragment', 'uncertain' sequences from the dataset. The dataset obtained from above described step was made non-redundant by using CD-HIT program with 40% threshold after which we got 13,271 sequences. After removing the protein sequences, which were also present as non-ERRPs in training dataset, we selected only those non-ERRPs, which had ER targeting consensus sequence [KRHQSA]-[DENQ]-E-L (*Raykhel et al., 2007*) (searched by using standalone version of ScanProsite (*Gattiker, Gasteiger & Bairoch, 2002*)) and got 2,900 non-ERRP sequences (Table 1; Table S3).

## Annotation of proteome

In order to show the real life usage and efficiency of our method, we annotated six proteomes namely *Homo sapiens, Mus musculus, Saccharomyces cervisiae, Caenorhabditis elegans, Drosophila melanogaster* and *Arabidopsis thaliana*. Their complete proteome set was downloaded from UniProt which had 68554, 45185, 5450, 26109, 22024 and 31527 proteins respectively.

## Support vector machine

SVM is a machine-learning algorithm to carry out pattern recognition and regression analysis on a given dataset (*Vapnik, 1995*). During training, SVM maps input data to the higher dimension and generates model, which can be used for prediction of an unknown example. In this work we used freely downloadable SVM_light package available at http://svmlight.joachims.org/ to implement SVM. During training, SVM needs only fixed-length feature along with their class as input. In the present work, fixed-length input feature of variable length protein sequence was obtained by amino acid composition, pseudo amino acid composition, dipeptide composition and different combinations of fragmented amino acid compositions.

## Cross-validation and performance evaluation

Cross-validation is a way to estimate the performance of a prediction model during training. During cross-validation, whole data is divided into two distinct sets; one set (called as training set) is used to train the model and second set (called as test set) is used for performance evaluation of model on a dataset that was not used during training. In prediction methods, three cross-validation approaches are most frequently used: independent dataset test, subsampling test (N-fold cross-validation) and jack-knife test or *leave-one-out* cross-validation (LOOCV). Among the three cross-validation approaches, LOOCV gives unique result for a given benchmark dataset and hence used in a number of prediction methods (*Kumar et al., 2014b*; *Kumar et al., 2015*; *Kumari, Kumar & Kumar, 2014*; *Lin et al., 2011*; *Xiao et al., 2013*; *Xu et al., 2013*). In the present study, we have used LOOCV approach for the evaluation purpose, in which all except one sequence of dataset is used as training set and remaining one sequence as test set. This process is repeated till each sequence has been used for testing. At a selected parameter, SVM was trained using the training set and predictive performance of model was evaluated on corresponding test

set. The prediction performance of trained model was calculated by averaging over all test set predictions.

To evaluate the performance of each trained SVM model, we used standard parameters regularly used in other prediction methods for prediction evaluation namely sensitivity, specificity, accuracy and Matthews Correlation Coefficient (MCC) (*Kumar, Gromiha & Raghava, 2008*; *Kumar et al., 2014a*; *Kumar et al., 2014b*; *Kumar et al., 2015*; *Panwar, Arora & Raghava, 2014*) as formulated below:

$$\text{Sensitivity} = \frac{TP}{TP+FN} \times 100 \tag{1}$$

$$\text{Specificity} = \frac{TN}{TN+FP} \times 100 \tag{2}$$

$$\text{Accuracy} = \frac{TP+TN}{TP+FP+TN+FN} \times 100 \tag{3}$$

$$\text{MCC} = \frac{(TP \times TN) - (FP \times FN)}{\sqrt{(TP+FP)(TP+FN)(TN+FP)(TN+FN)}} \tag{4}$$

where TP represents true positive (proteins, which are actually ERRPs and were also predicted as ERRPs), TN represents true negative (proteins, which are actually non-ERRPs and also predicted as non-ERRPs), FP represents false positive (the number of non-ERRP predicted as ERRPs), FN represents false negative (number of proteins, which are actually ERRPs but predicted as non-ERRPs).

## Sequence derived features

As SVM needs fix length input to operate, hence we converted variable length protein sequence into fixed length vector. Amino acid, pseudo amino acid, dipeptide, and split amino acid compositions were used to encapsulate the protein sequence information into fixed length vector.

## Amino acid composition

The amino acid composition is the fraction of each amino acid within a protein sequence. It has been used extensively in past for prediction of different protein features (*Kumar, Gromiha & Raghava, 2007*; *Kumar, Gromiha & Raghava, 2011*; *Kumar et al., 2014a*; *Kumar et al., 2015*; *Wang, Xiao & Chou, 2011*). The fraction of each amino acid was calculated by using the following formula:

$$\text{Acomp}(i) = \frac{\text{Total number of an amino acid}(i)}{\text{Total number of amino acid in protein}(p)} \times 100 \tag{5}$$

where Acomp(i) is the percentage of amino acid 'i' in a given protein and (i) can be any amino acid of a particular protein (p).

## Pseudo amino acid composition

One of the major drawbacks of the amino acid composition is the loss of information about amino acid sequence order from the input features. To deal with this problem Chou proposed a new way of amino acid composition called as pseudo-amino acid composition (*Chou, 2001*). Pseudo-amino acid composition of a protein is actually a set of discrete

numbers, which is derived from its amino acid sequence. It allows representation of both compositional and positional amino acid pattern in a discrete mode (*Limongelli, Marini & Bellazzi, 2015*). It also reflects sequence order information and length of the proteins (*Chou, 2005*). Many tools (*Du et al., 2012*; *Han, Yu & Anh, 2014*; *Kumar et al., 2015*; *Liu et al., 2014*; *Mondal & Pai, 2014*) are available to calculate the pseudo-amino acid composition of protein sequences. Here we used PseAAC-Builder (*Du et al., 2012*) at default parameters (weight factor = 0.05 and Lambda parameter = 1).

## Dipeptide composition

Dipeptide composition is a modified form of amino acid composition, which has been used in a number of protein classification methods like nuclear receptor family classification (*Bhasin & Raghava, 2004*; *Kumar et al., 2014b*), membrane protein prediction, ion channels prediction (*Lin & Ding, 2011*), protein fold prediction (*Shamim, Anwaruddin & Nagarajaram, 2007*), heat shock protein and its class prediction (*Kumar, Gromiha & Raghava, 2011*; *Kumar, Kumari & Kumar, 2016*; *Reczko & Bohr, 1994*). Dipeptide composition calculates the number of all possible types of amino acid pairs in a protein sequence in a sliding window mode with step size 1 and window size 2. Hence, it incorporates amino acid composition along with the local order information also. Dipeptide composition describes each protein sequence in form of 400-dimension feature vectors, which can be calculated as

$$\mathrm{Dipep(n)} = \frac{\text{Total number of Dipep(n)}}{\text{Total number of all possible dipeptide (p)}} \times 100 \qquad (6)$$

where Dipep(n) is the percentage of dipeptide 'n' in a given protein and (n) can be any dipeptides out of the all possible 400 dipeptides of a particular protein (p).

## Split amino acid composition

Secretory proteins are known to contain signal to guide them outside of the cell. This feature also distinguishes them from the intracellular proteins (*Wrzeszczynski & Rost, 2004*). Hence sequence encapsulation, which can highlight this difference, ought to be more informative. In view of this, we divided each protein sequence into different non-overlapping fragments, viz. 2-parts and 3-parts, and calculated amino acid composition of each part separately. Fragment that contains the signal sequence would have amino acid compositions different from the remaining fragments, thereby highlighting the presence of signal sequence(s). Hence split amino acid composition might provide more realistic information than amino acid and dipeptide compositions of whole protein. This approach has been used earlier to encode input protein features (*Afridi, Khan & Lee, 2012*; *Kumar & Raghava, 2009*; *Kumar, Verma & Raghava, 2006*; *Verma, Varshney & Raghava, 2010*; *Hayat, Khan & Yeasin, 2012*). In this work, we used 2-parts and 3-parts-split amino acid compositions as input features.

In 2-parts-split amino acid (SAAC-2-parts), first we divided each sequence in two different ways. We named these two split sets as N-terminal SAAC (N-ter-SAAC) and C-terminal SAAC (C-ter-SAAC) respectively. N-ter-SAAC had protein sequences in two parts, (i) 25 amino acids of N-terminal and (ii) the remaining residues of the sequence.

Similarly, C-ter-SAAC included (i) 25 amino acids of C-terminal and (ii) the remaining sequence amino acids. For both N-ter-SAAC and C-ter-SAAC, we calculated the amino acid composition of each part separately.

In 3-parts-split amino acid composition (SAAC-3-parts), protein sequence was divided into three parts: N-terminal, middle part and C-terminal and then amino acid composition of each part was calculated separately. Figure 1 depicts a protein sequence K, which is divided into three parts; $K_N$, $K_C$ and $K_R$ where $K_N$ is the 25 N-terminal residues, $K_C$ represents the 25 C-terminal residues and $K_R$ represents remaining sequence. Calculation of amino acid composition of each part provided 20 vectors, so the length of final input vector was 60 for each protein. Due to the fractional nature of amino acid composition, it is likely to highlight the enrichment and depletion of specific amino acids in each segment and reveal the biasness in distribution of amino acids in a protein.

## RESULTS

### Enrichment and depletion pattern of amino acids in ERRPs

The amino acid composition of proteins of each subcellular location is specifically adopted to suit its surrounding environment (*Andrade, O'Donoghue & Rost, 1998*). In order to understand this pattern in case of ERRPs, we analysed the enrichment and depletion pattern of amino acids using Composition Profiler (*Vacic et al., 2007*). Composition Profiler detects the enrichment and depletion patterns of individual amino acid as well as group of amino acids classified on the basis of different physico-chemical and structural properties by using two groups of protein sequences as query and background sample. The fractional difference (in terms of enrichment and depletion) between distribution of a particular amino acid in query (*d1*) and background dataset (*d2*) is calculated as follows:

$$\text{Compositional difference}(D) = \frac{d1 - d2}{d2}. \tag{7}$$

In this work, during enrichment and depletion analysis, all ERRPs were merged into a single group as a query sample while all non-ERRPs were used as background sample. As per Composition Profiler analysis, at *P*-value $\leq 0.05$, aromatic (phenylalanine, tyrosine, tryptophan), negatively charged (aspartic acid and glutamic acid), polar (tyrosine) and hydrophobic (valine, leucine, phenylalanine, tryptophan) residues were enriched while positively charged (arginine) residues were depleted in ERRPs (Fig. 2). It has been observed that ER proteins are rich in transmembrane domain (*Schuldiner & Weissman, 2013*; *Scott et al., 2004*). The enrichment of aromatic and hydrophobic residues in ERRPs is in accordance with this observation.

### Composition based SVM modules
#### *Amino acid, pseudo amino acid and dipeptide composition based SVM modules*

First, we used simple amino acid composition as SVM input. Using LOOCV approach of training, the maximum accuracy we obtained was 73.34% (MCC = 0.29). When information of amino acids arrangement order was provided in the form of pseudo amino

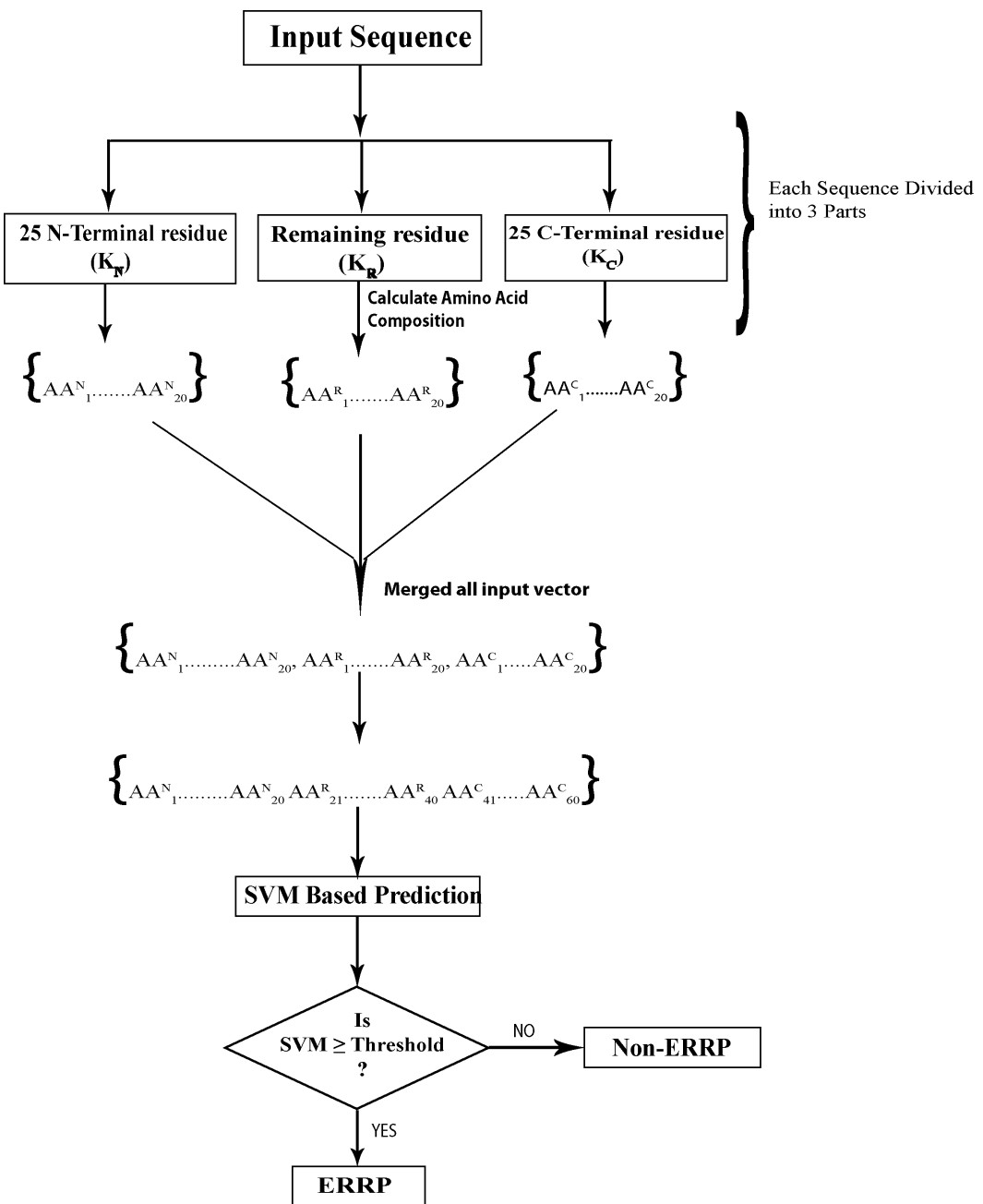

**Figure 1** Prediction schema of Endoplasmic reticulum resident proteins using split amino acid based input to SVM.

acid composition based input, prediction accuracy increased to 74.85% (MCC = 0.30). When dipeptide composition was used as SVM input, we found a lower performance and the maximum accuracy decreased to 72.28% with MCC value 0.26 (Table 2). These results show that pseudo-amino acid composition based SVM module is better than the simple amino acid and dipeptide composition based modules.

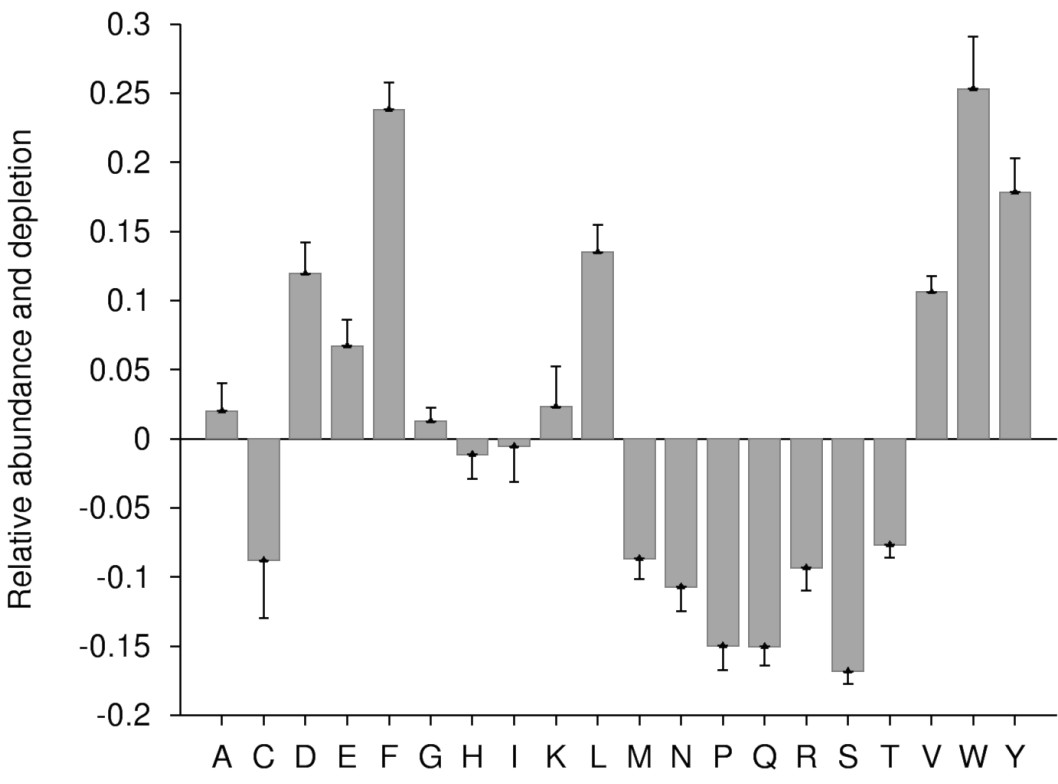

**Figure 2** **Relative enrichment and depletion profile of amino acids in ERRPs with reference to non-ERRPs.** A negative value indicates the depletion and a positive value indicates the enrichment of amino acid.

**Table 2** **Performance of SVM-Models based on different input vectors during *leave-one-out* cross validation.**

| Input vector | Sensitivity | Specificity | Accuracy | MCC | AUC |
|---|---|---|---|---|---|
| AAC | 72.58 | 73.42 | 73.34 | 0.29 | 0.78 |
| Pseudo AAC | 70.97 | 75.25 | 74.85 | 0.30 | 0.77 |
| Dipeptide Composition | 69.35 | 72.58 | 72.28 | 0.26 | 0.76 |
| N-ter-SAAC | 75.00 | 80.00 | 79.53 | 0.37 | 0.83 |
| C-ter-SAAC | 72.58 | 70.92 | 71.07 | 0.27 | 0.77 |
| SAAC-3 parts | 79.84 | 81.58 | 81.42 | 0.42 | 0.85 |

**Notes.**
AAC: amino acid composition, Pseudo AAC: pseudo amino acid composition, N-ter-SAAC: 25 N-terminal and remaining sequence composition, C-ter-SAAC: 25 C-terminal and remaining sequence composition, and SAAC-3 parts: 25 N-terminal, 25 C-terminal and remaining amino acid composition. MCC and AUC represent Matthews's correlation coefficient and area under ROC curve, respectively.

### Split amino acid composition based SVM modules

In SAAC based SVM modules, we used three input features to train SVM. With N-ter-SAAC, we found maximum accuracy 79.53% with MCC value 0.37. With C-ter-SAAC the accuracy reduced to 71.07% and MCC value to 0.27. But with SAAC-3-parts, the accuracy increased to 81.42% with MCC value 0.42 (Table 2).

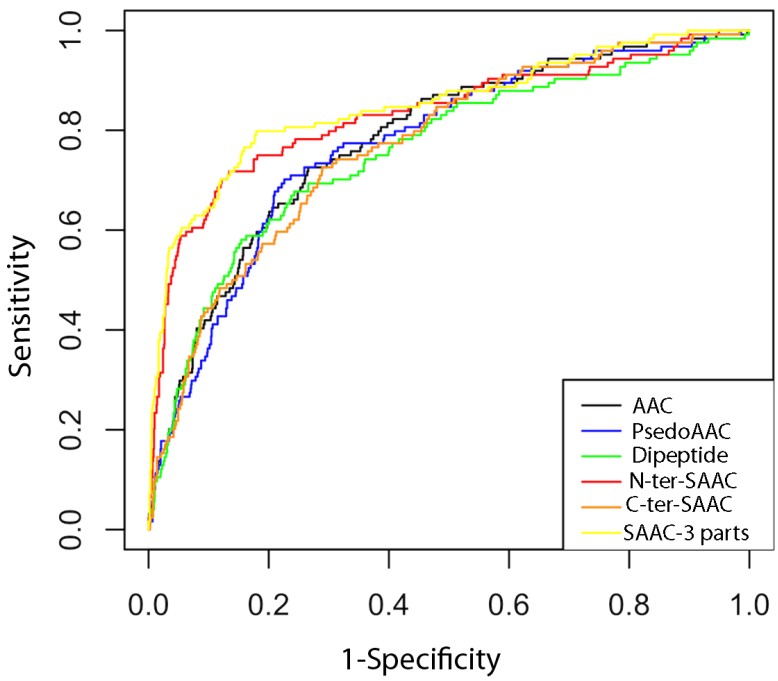

**Figure 3** **ROC plots of ERRPs prediction using different amino acid features.** AAC, PsedoAAC, Dipeptide, N-ter-SAAC, C-ter-SAAC and SAAC-3-parts represents amino acid composition, pseudo-amino acid composition, dipeptide composition, 25 N-terminal and remaining amino acid composition, 25 C-terminal and remaining amino acid composition and 25 N-terminal, 25 C-terminal and remaining amino acid composition, respectively.

The performance obtained from different SVM models can be explained by the fact that retention of proteins inside ER is mediated by some signal sequences or related sequences (*Gomord, Wee & Faye, 1999*). Hence, an input vector which resolves the signal sequences efficiently is expected to perform better than others. We feel that Since SAAC-3-parts divide a protein into three parts hence, it can resolve the signal motif better, therefore, SAAC-3-parts based SVM model has the maximum accuracy.

## Receiver operating characteristics curve and area under curve analysis

Receiver Operating Characteristics (ROC) curve is a graphical way to illustrate the performance of a classifier. It is plotted as 'sensitivity' *vs.* '1-specificity' and shows trade-off between true and false positives (*Fawcett, 2006*; *Hajian-Tilaki, 2013*). The area under the ROC curve is called AUC value (*Bradley, 1997*) which can be used to quantify the prediction performance of a classifier. Higher the AUC value better is the prediction. In this study, we used ROCR package (*Sing et al., 2005*) for plotting the ROC curve and finding AUC value. As shown in Fig. 3 and Table 2, the ROC curves and the AUC values also confirms that the classifier based on SAAC-3 parts is better than classifiers developed using other sequence encapsulations.

**Table 3** Comparative performance of ERPred vis-à-vis iLoc-Euk, Cello v.2.5 and Euk-mPloc 2.0 on independent dataset.

| Methods | Sensitivity (%) | Specificity (%) |
|---|---|---|
| ERPred | 72.31 | 83.69 |
| Cello 2.5 | 16.92 | 99.86 |
| iLoc-Euk | 15.38 | 99.76 |
| Euk-mPLoc 2.0 | 66.15 | 99.00 |

## Comparison and evaluation of other prediction approaches

ERRPs follow two basic mechanisms to stay inside the ER: (i) signal dependent and (ii) signal independent. Signal dependent proteins have ER-retention signals while signal independent proteins do not have these signals, e.g., some cereal prolamin storage proteins (*Sophie Pagny, Faye & Gomord, 1999*). In order to find out the efficiency of signal based approach, we evaluated ER retention signal based prediction of ERRPs using Prosite motif database (*Hulo et al., 2006*). Using standalone version of ScanProsite (*Gattiker, Gasteiger & Bairoch, 2002*), out of 124 proteins of training dataset, we were able to find ER retention signal ([KRHQSA]-[DENQ]-E-L) in only 66 proteins, which shows that signal sequence is not present in all ERRPs. This shows that signal based approach may not be appropriate for complete ERRP repertoire prediction of any proteome. Similarly there are many non-ERRPs, which also have ER-retention signal but they do not reside into the ER. Efficiency of signal based approach also suggests that presence of ER-retention signal may not be the only factor responsible to hold a protein inside ER. It also implies that a good ER predictor should also be able to discriminate and recognize ERRPs that does not have ER-retention signals and the non-ERRPs that have ER-signals. So, in order to evaluate our method on the above-mentioned two parameters, we analysed 65 ERRPs of independent dataset for presence of ER-retention signal by ScanProsite. We found that only 21 proteins had ER-signal while remaining 44 proteins did not have any ER-signal. Further, when we predicted whether these 44 ERRPs localizes to ER or not using ERPred, 27 proteins were predicted as ERRPs and 17 as non-ERRPs. We also benchmarked the ERPred using 2,900 non-ERRPs of independent dataset containing ER-retention signal sequences. ERPred correctly predicted the non-ERRPs with 83.69% specificity (Table 3). This shows that the mere presence or absence of ER-retention signal does not necessarily implies ERRP or non-ERRP respectively. *Wrzeszczynski & Rost (2004)* also observed that most of the known ER-retention signals are too specific and/or too inaccurate to be used in alone for automatic protein annotation. Their observation was also based on experimentally characterized short sequence motifs and sequence similarity search to experimentally characterized proteins. They also showed that either >80% pairwise sequence identity or $<10^{-100}$ $E$-value can be used for homology-transfer based annotation and at very low coverage which severely limit the number of true positives.

To annotate novel proteins, 'homology based function transfer' is the most popular approach. This approach involves BLASTing novel gene/protein against a gene and/or protein database to find well-annotated homolog(s) and transfer the annotation of top hit(s) to the query proteins (*Brown, Krishnamurthy & Sjolander, 2007*; *Radivojac et al.,*
*2013*; *Rost et al., 2003*). Since similarity search considers whole protein sequence rather than a short stretch of ER-retention signal, this approach should be more effective than the signal based approach. In order to assess the efficiency of BLAST to discover new ERRPs, we evaluated the performance of BLAST search using the proteins of training dataset. In this process, each protein out of 124 was used as query and remaining 123 proteins as database. At $e$-value threshold $\geq 1e-3$, only 56 proteins found ERRP as first BLAST hit. Remaining 68 proteins did not get any hit. It shows that finding homologous sequence using BLAST is also not sufficient to search all ERRPs. Our inference regarding usage of BLAST in prediction of ERRPs is also supported by the observation noted by *Wrzeszczynski & Rost (2004)* that even very high levels of sequence similarity might not be sufficient to infer ER localization without error.

Overall, the results show that performance of ERPred does not depend either on ER retention signal or sequence homology. Therefore, we feel our approach can be an appropriate alternative to the annotation transfer on the basis of homologous protein and/or signal sequence based approaches for high throughput proteome annotation.

## Evaluation of ERPred on independent dataset

To derive unbiased results we evaluated proposed prediction schema using proteins of independent dataset. The outcome of prediction was used to assess the sensitivity and specificity of ERPred. We found ERPred showed sensitivity 72.31% and specificity 83.69% on independent dataset (Table 3). This shows that our method not only detects ERRPs with high sensitivity but also recognizes non-ERRPs with high specificity.

## Comparison with other methods

To the best of our knowledge, there is no existing method which can specifically predict ERRPs. Though a few subcellular localization prediction methods considered ER as one of many locations in the cell these predict ER proteins along with proteins of other subcellular locations like, ProLoc-GO (*Huang et al., 2008*), KnowPredsite (*Lin et al., 2009*), SLocX (*Ryngajllo et al., 2011*), iLoc-Animal (*Lin et al., 2013*), Cello v.2.5 (*Yu et al., 2006*), Euk-mPloc (*Chou & Shen, 2007*), Euk-mPloc 2.0 (*Chou & Shen, 2010*), HybridGO-Loc (*Wan, Mak & Kung, 2014*), mGOASVM (*Wan, Mak & Kung, 2012*), Hum-Ploc (*Chou & Shen, 2006*) and iLoc-Euk (*Chou, Wu & Xiao, 2011*). Unfortunately, ProLoc-GO (*Huang et al., 2008*), KnowPredsite (*Lin et al., 2009*) were not found in a working state and HybridGO-Loc and mGOASVM considered only viral and plant proteins. Hum-Ploc is meant only for human proteins, SLocX (*Ryngajllo et al., 2011*) predicts only ER proteins of *Arabidopsis thaliana* and iLoc-Animal (*Lin et al., 2013*) predicts only proteins of animal system. Since ERPred prediction is not organism/taxa specific hence we compared our method with those methods which are not restricted to a particular organism or plant/animal like Cello v.2.5 (*Yu et al., 2006*), iLoc-Euk (*Chou, Wu & Xiao, 2011*) and Euk-mPLoc 2.0 (*Chou & Shen, 2010*) (an updated version of Euk-mPloc (*Chou & Shen, 2007*)) using independent dataset. The sensitivity of ERPred on independent dataset was 72.31% with 83.69% specificity while for Cello v.2.5 the corresponding values were 16.92% and 99.86% respectively and for iLoc-Euk the sensitivity and specificity was 15.38% and 99.76% respectively (Table 3).
**Table 4  Proteome level prediction of ERRPs using ERPred and comparison with ER-GolgiDB and Locate databases.**

| Proteome | Number of proteins in complete proteome | Number of ERRPs predicted by ERPred | % of ERRPs in proteome | Number of protein in different database | |
|---|---|---|---|---|---|
| | | | | ER-GolgiDB | Locate |
| *H. sapiens* | 68,554 | 2,293 | 3.34 | 2,543 | 1,762 |
| *M. musculus* | 45,185 | 1,781 | 3.94 | 2,248 | 1,588 |
| *D. melanogaster* | 22,024 | 707 | 3.21 | 1,075 | – |
| *C. elegans* | 26,109 | 1,014 | 3.88 | 1,196 | – |
| *S. cerevisiae* | 5,450 | 148 | 2.72 | 407 | – |
| *A. thaliana* | 31,527 | 1,089 | 3.45 | 1,765 | – |

Hence Euk-mPLoc showed better performance than iLoc-Euk and Cello v 2.5 in terms of sensitivity, which was 66.15%. As shown in Table 3, among all the predictors, ERPred performed better in prediction of ERRPs but its performance in terms of specificity or non-ERRPs was lesser than other predictors. It might be due to the fact that all predictors, which we have used here for evaluation on the basis of independent dataset, were intended to predict multiple locations. The result shows that in majority of cases they were predicting locations other than ER, due to which their overall efficiency was higher. In other words, in most cases, they were predicting ERRPs and non-ERRPs as non-ERRPs, which increased their performance. But, if we carefully analyse the results, it can be observed that their performance is not equally good for the prediction of ERRPs. This also vindicates our attempt to develop a dedicated predictor for ERRPs.

We also evaluated the performance of BLAST based annotation using sequences of training and independent dataset as database and query respectively. We found that only 40 ERRP sequences of independent dataset were annotated as ERRP. Remaining 25 sequences did not get any hit.

## Annotation of proteome

We selected six different proteomes ranging from less complex yeast to the most complex human for the purpose of annotation. In *H. sapiens*, out of 68,554 proteins, ERPred predicted 2,293 proteins as ERRP, which is 3.34% of the proteome. In case of *M. musculus,* ERPred predicted 1,781 ERRPs out of 45,185, which is 3.94% of the proteome. In *D. melanogaster,* ERPred predicted 707 proteins as ERRPs out of total of 22,024 proteins, which is 3.21% of the proteome. In case of *C. elegans*, ERPred predicted 1,014 proteins as ERRPs out of 26,109 proteins, which is 3.88% of the proteome and 148 proteins are ERRPs out of 5,450 proteins, which is 2.72% of the *S. cerevisiae* proteome. In *A. thaliana*, ERPred predicted 1,089 proteins as ERRPs (3.45% of proteome) out of 31,527 proteins (Table 4).

Since there is no method which can predict ERRP at proteome level so we compared our prediction method with two regularly updated existing databases ER-GolgiDB (*Wrzeszczynski & Rost, 2004*), the database which contained all the proteome which we had annotated and Locate (*Sprenger et al., 2008*), the database containing only *H. sapiens* and *M. musculus* proteome. ER-GolgiDB contains information about ER and Golgi localization based on sequence homology to experimentally annotated proteins. Locate

contains data about subcellular localization of proteins from the RIKEN FANTOM4 mouse and human protein sequence set. The information content of Locate was determined by a high-throughput, immunofluorescence-based assay and from peer-reviewed publications.

The ERPred estimated the existence of 2,293 ERRPs in human, representing about 3.34% of human proteome. The estimated number was close to the 2,543 and 1,762 proteins of ER-GolgiDB and Locate databases respectively. In mouse, ERPred estimated 1,781 proteins (3.94% of the proteome), which was in line with the estimation of 1,588 by Locate and slightly lower than ER-GolgiDB databases respectively. Nearly identical fraction of ERRPs in mouse and human proteomes was expected due to their close evolutionary relationship. We feel the difference in numbers might be due to the proteome size. We expect the number will increase, as more and more proteins will be added to the proteome. In case of *C. elegans* proteomes, the fractions of ERRPs was almost equal while in *D. melanogaster* and *A. thaliana* the fraction was slightly less. This might be due to the fact that both ER-GolgiDB and Locate were last updated in 2010 and 2008 respectively.

## WEBSERVER AND STANDALONE SOFTWARE

Based on the schema described above, we have established a webserver ERPred, which is freely available at http://proteininformatics.org/mkumar/erpred/index.html. This webserver allows users to predict ER resident proteins. User can submit protein sequences in FASTA format for prediction. The maximum limit of prediction is 25 sequences at a time. If a user submits more than 25 sequences, only the first 25 sequences will be processed for prediction. For batch prediction, we have also made a standalone version of this software, which is available at download page of the webserver. ERPred also provides option to select different threshold values, which can be used to select the level of false and true positive predictions.

## DISCUSSION

Even though the importance of endoplasmic reticulum resident proteins has been established undoubtedly, only a few methods have considered ER as a separate subcellular locality. In these methods, the accuracy of prediction of ERRPs is very less in comparison of other locations. The purpose of the present study is therefore to describe a tool, ERPred, for the prediction of ERRPs. The tool uses only the protein sequence information for prediction. While developing the method, we tried different forms of sequence compositions; namely amino acid, dipeptide, and pseudo amino acid compositions. Better performance of a pseudo-amino acid composition based SVM module than the simple amino acid and dipeptide composition based modules reveals that pseudo amino acid composition encapsulates amino acid ordering information better than dipeptide composition. But the best performance was obtained when amino acid compositions of different part of proteins were used. The amino acid composition based SVM module gave 73.34% accuracy. The performance was decreased to ∼1% giving 72.28% accuracy when we used dipeptide composition as input. Dipeptide composition was used to incorporate composition as well as local order information. Further, when we used pseudo-amino acid composition, it

showed a better accuracy of 74.85%. In order to include information of enrichment and depletion of amino acids in a specific region of ERRPs, we also used SAAC in which a protein sequence was divided into three different ways (N-ter-SAAC, C-ter-SAAC, and SAAC-3 parts). N-ter-SAAC contained two sets of amino acid compositions, one from 25 N-terminal residues and other from rest of the residues while C-ter-SAAC also had two sets of amino acid compositions firstly from 25 C-terminal residues and remaining residues. SAAC-3 parts had three sets of vectors from 25 N-terminal residues, 25 C-terminal residues and the remaining sequence residues. SVM modules based on N-ter-SAAC, C-ter-SAAC and SAAC-3 parts respectively achieved accuracy of 79.53%, 71.07% and 81.42%. This shows that the SVM module developed from C-ter-SAAC was least efficient whereas the SAAC-3 parts based module was better for discriminating ERRPs from non-ERRPs. The performance obtained from different SVM models can be explained by the fact that retention of proteins inside ER is mediated by some signal sequences or related sequences (*Gomord, Wee & Faye, 1999*). Hence an input vector, which resolves the signal sequences efficiently, is expected to perform better than others.

With use of machine learning techniques, we explored which form of composition can best help in high-level successful prediction of ERRPs. The results show that split composition-based features were most predictive. KDEL and its variants is the most common signal present generally at the C-terminal of ERRPs. The other well-characterized retention signal is the di-lysine and di-arginine motifs present at C- and N-terminals respectively. But these signals are neither present in all ERRPs nor sufficient to retain all to ER (*Gao et al., 2014*; *Ma & Goldberg, 2013*). This also suggests that other parts of proteins also play a very important role in retention of ERRPs in ER (*Scott et al., 2004*). The most surprising result we obtained was the performance of dipeptide composition based SVM models, which has shown least performance among all. Ideally it should have picked up the signals responsible for retention of ERRPs in ER and should have shown better performance than at least amino acid and pseudo amino acid compositions based models. The higher performance of SAAC-3 parts in comparison to the C- and N-ter SAAC can also be explained in light of this fact that in SAAC-3 parts the presence of signal as well as other unknown factors present at non-terminal regions can be highlighted better than the C- and N-ter SAAC.

We also benchmarked the performance of our method vis-à-vis other predictors using an independent dataset. The results showed that ERPred is better than other tools. We also observed that the performance obtained was roughly equal to what we have obtained earlier during the cross-validation.

## CONCLUSION

ER is the one of the most important cellular organelles of eukaryotic cells. Knowledge of complete ER resident protein repertoire will help in better understanding of biological pathways of ER. In the present study, a sequence-based tool for prediction of ERRPs has been reported for the first time. We have also developed an open access web-server and standalone software for batch mode prediction of ERRPs and it can be accessed at

http://proteininformatics.org/mkumar/erpred/. Independent assessment with non-specific sub-cellular localization methods showed the higher efficiency and also justified the need of an ERRP specific prediction method. We expect that this tool will bring new insights about understanding of the functions of ER.

### Funding

This work was funded by the Science & Engineering Research Board (SERB), the Department of Science & Technology, Government of India under Fast Track Scheme for Young Scientist (Grant no. SR/FT/LS-84/2010) and a University Grant Commission Major Research Project Grant (41-38/2012(SR)) to Manish Kumar. Ravindra Kumar is supported as a Senior Research fellow under a grant (number 20-12/2009(ii)EU-IV) from the University Grant Commission India. Bandana Kumari is a Senior Research fellow and is supported by grant (number BIC/11(33)/2014) funded by the Indian Council of Medical Research, India. The funders had no role in study design, data collection and analysis, decision to publish, or preparation of the manuscript.

### Grant Disclosures

The following grant information was disclosed by the authors:
Science & Engineering Research Board (SERB).
Department of Science & Technology, Government of India under Fast Track Scheme for Young Scientist: SR/FT/LS-84/2010.
University Grant Commission Major Research Project: 41-38/2012(SR).
University Grant Commission India: 20-12/2009(ii)EU-IV.
Indian Council of Medical Research, India: BIC/11(33)/2014.

### Competing Interests

The authors declare there are no competing interests.

### Author Contributions

- Ravindra Kumar conceived and designed the experiments, performed the experiments, analyzed the data, wrote the paper, prepared figures and/or tables, reviewed drafts of the paper.
- Bandana Kumari analyzed the data, wrote the paper, reviewed drafts of the paper.
- Manish Kumar conceived and designed the experiments, analyzed the data, wrote the paper, reviewed drafts of the paper.

### Data Availability

The raw data has been supplied as Supplementary Material.
ERPred can be accessed at http://proteininformatics.org/mkumar/erpred/download.html.
## Supplemental Information

Supplemental information for this article can be found online at http://dx.doi.org/10.7717/peerj.3561#supplemental-information.

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
