# Peer review of "Prediction of endoplasmic reticulum resident proteins using fragmented amino acid composition and support vector machine"

_PeerJ, doi:10.7717/peerj.3561_

## Round 0.1 · original submission · Minor Revisions

In particular, please could you address the comments raised by reviewer 2, who would like you to comment on and address a number of issues linked to KKXX motifs and their implications for your classifier, and the nature of the potential similarity between testing and training datasets (40% seems quite a high cutoff to use?). The other comments should also be addressed, and you might like to seek improvements to the written English as noted (and throughout the manuscript).

Reviewer 1 ·

Basic reporting

no comment.

Experimental design

no comment.

Validity of the findings

no comment.

Additional comments

The paper titled “Prediction of endoplasmic reticulum resident proteins using fragmented amino acid composition and support vector machine” reported a computational method to identify ER-resident proteins from sequences. As far as I can tell, this is the first study to explore computational methods in identifying ER-resident proteins. The manuscript is written well. The results is very promising and informative for future studies. I have only one minor comment: Recently, several studies explorer the methods in predicting Golgi-resident proteins, if the authors wish to, they may mention these contributions in their paper, which may improve the value of this work much.

Reviewer 2 ·

Basic reporting

This is an effectively written manuscript for the mort part, well described and illustrated with Figures and Tables. There are relatively minor issues with the English that could be improved, e.g. the final sentence in the Abstract starts "But ...".

The hypothesis is clear, that amino acid sequence-based analysis can improve the prediction of ER-resident proteins, beyond effectively a simple first order 'KDEL' approach. This is demonstrated, in part (see general comments section).

Experimental design

Experimental design is good for the most part, with a couple of issues that I will raise in General comments.

Validity of the findings

Again, the procedures are effective, so that findings are likely to be valid, with a couple of caveats that I will address in General comments.

Additional comments

I find it easier to summarise comments here, allowing for a prioritisation of issues (most important first).

* In the Introduction and with ramification in the Results and Discussion, it is not clear to me whether the authors consider the existing literature as going beyond the KDEL ER retention signal (or the related ProSite pattern that they use). There exists work e.g. on DHHC proteins and others(https://www.ncbi.nlm.nih.gov/pubmed/23481256, https://www.ncbi.nlm.nih.gov/pubmed/24794130) that consider KKXX and other motifs as ER retention signals rather than ER transit signals. Could the authors discuss this work, and whether there is any evidence that their dipeptide composition features could be picking up such motifs?

* With a 40% sequence id threshold for proteins within and between test and training sets, there still remains the possibility of annotating the test set by homology with the training set. There is some discussion of BLAST analysis within the training set, but what about between training and test sets?

* The authors give numbers for prosite KDEL-like motif annotation of the 124 protein set, what about the other sets?

* A balance of 1200 to 124 for negative vs positive ER-retained protein training sets is quoted as being sufficient, in the authors 'view'. It probably is OK, and probably is handled the correct way, but a reference could be given here.

* As with all machine learning methods, it would be nice to have more interpretation than is perhaps possible, concerning the features that contribute to prediction accuracy. Any further discussion here would be welcomed, e.g. relating to KK-containing motifs, as mentioned previously.

Overall, I think this is a nice piece of work, my comments are inquisitive, rather than particular critical.

---

## Round 0.2 · accepted · Accept

From the Editor: Congratulations on your paper being accepted. As a final request, could you make some small edits to the abstract? As written it could be misinterpreted slightly. Your prediction tool makes predictions, it doesn't "find" ER retained proteins so could you edit the appropriate sentence to read something like:
"We have also annotated six different proteomes to predict the candidate endoplasmic reticulum resident proteins in them"

Reviewer 2 ·

Basic reporting

OK

Experimental design

OK

Validity of the findings

OK

Additional comments

The authors have addressed have addressed sufficiently the comments that I made on their original submission.